# Non-Pharmacological Management of Gestational Diabetes Mellitus with a High Fasting Glycemic Parameter: A Hospital-Based Study in Vietnam

**DOI:** 10.3390/jcm13195895

**Published:** 2024-10-02

**Authors:** Hang Giang Nguyen, Khanh Trang Huynh Nguyen, Phuc Nhon Nguyen

**Affiliations:** 1Department of Obstetrics and Gynecology, Pham Ngoc Thach University of Medicine, Ho Chi Minh City 730000, Vietnam; duyquang1316@gmail.com (H.G.N.); tranghnk08@gmail.com (K.T.H.N.); 2Hung Vuong Hospital, Ho Chi Minh City 700000, Vietnam; 3Department of High-Risk Pregnancy, Tu Du Hospital, Ho Chi Minh City 71012, Vietnam; 4Tu Du Clinical Research Unit (TD-CRU), Tu Du Hospital, Ho Chi Minh City 71012, Vietnam

**Keywords:** diet therapy, gestational diabetes mellitus, OGTT, physical activities, PPAQs

## Abstract

**Background/Objectives**: The prevalence of gestational diabetes mellitus (GDM) is increasing at an alarming rate worldwide. Delayed management can lead to adverse composite outcomes for both mother and her offspring. To our knowledge, the clinical association between glycemic parameters and the results of the non-pharmacological GDM approach remains limited; thus, this study aimed to address this important clinical issue in the literature. **Methods**: This was a retrospective cross-sectional study of 174 Vietnamese pregnant women with the positive oral glucose tolerance test (OGTT) for a high fasting glycemic parameter at Hung Vuong Hospital from 04/2022 to 07/2022. This study aimed to evaluate the success rate of GDM with an elevated index of fasting glycemic concentration which was managed after 2 weeks of a dietary regimen combined with adequate physical activities and to reveal its related factors. **Results:** Out of 174 singleton pregnancies that met the inclusion criteria, 103 GDM pregnant women were successfully managed after 2 weeks of monitoring (59.2%; 95% confidence intervals (CI): 51.9–66.5). The study revealed a fair correlation between the corresponding test of blood glucose at OGTT and after 2 weeks of GDM management (*r* = 0.270–0.290, *p* < 0.0001). The GDM pregnant women with an elevated fasting glycemic parameter and with any of elevated 1 h or 2 h blood glucose levels and in cases of three elevated glycemic parameters (fasting, 1 h, and 2 h blood glucose at the initial results of OGTT) reduced the success rate of glycemic control to 56.5%, 49.2%, respectively, compared to the group with solely a high fasting index of blood glucose (69.6%). The pregnant women who participated in high-intensity sports activities related to a two-fold increase in success rate compared with the group engaging in light and moderate-intensity physical activity. **Conclusions**: The success rate of glycemic control in GDM women was highest in the group with solitary fasting hyperglycemia and lower in the contributory groups with two and three high parameters. The application of diet therapy plus physical activities among GDM pregnant women is potentially necessary to improve the effectiveness of treatment, minimize adverse pregnancy outcomes, and reduce substantially the hospitalization rate.

## 1. Introduction

Gestational diabetes mellitus (GDM) is defined as a disorder of glycemia occurring in pregnancy, a transitory form of pregnancy-induced diabetes. The rate of GDM is currently rising due to lifestyle modification in the new era, up to 16.2% of live births worldwide [1]. Particularly, the development of GDM pregnancy is an alert in low-middle income countries (LMICs) where assessment with the healthcare system remains limited, increasing to 20–25% in Southeast Asia [1,2]. The choice of criterion with different measurements can affect diabetes prevalence and the diabetes could be underestimated and underdiagnosed depending on available resources [3,4]. In a randomized clinical trial of gestational diabetes screening of 23,792 women, Hillier et al. found that the positive test of GDM with the one-step approach was greater than with the two-step approach (16.5% vs. 8.5%) [5]. According to several studies on Vietnamese pregnant women, the prevalence of GDM pregnancy varies following the diagnostic criteria, including 6.4% by The American Diabetes Association (ADA) and 20.0–22.8% by the International Association of Diabetes and Pregnancy Study Groups (IADPSG) and the World Health Organization (WHO) [6,7]. Recently, the clinical utilities of continuous glucose monitoring (CGM), point-of-care (POC) capillary blood glucose (CBG) testing, and point-of-care hemoglobin A1C (HbA1C) have been recently assessed for the management of GDM. However, these modalities have not been applied widely in clinical practice due to limited resource settings [8,9,10].

Accordingly, gestational diabetes mellitus impacts negatively pregnant women [11]. The uncontrolled hyperglycemia with an inappropriate diet could lead considerably to many adverse perinatal outcomes in term of ketosis, macrosomia, newborn birth injury, shoulder dystocia, congenital defects, preterm birth, maternal birth trauma, and newborn hypoglycemia [12,13,14]. Particularly, pre-pregnancy diabetes mellitus contributes to adverse pregnancy outcomes [15]. Furthermore, unfavorable long-term outcomes in association with metabolic disorders in women and infants have been reported [16,17]. The underlying mechanism is maternal hyperglycemia and hyperinsulinemia leading to increased insulin resistance, resulting in alterations and overnutrition in the fetus. These changes also contribute adversely to the offspring’s outcomes after birth [18,19,20]. Moreover, other issues associated with anxiety, depression, and poor mental health of pregnant women raise a concern in most parts of the world [21,22].

However, 70% of GDM could be successfully achieved by adequate dietary therapy and physical activities [23]. Preventing excessive fetal growth is a necessary treatment in the underlying management of GDM [24,25,26]. Pharmacotherapies including medicaments and insulin administration are only indicated when the target blood glucose level is not achieved with diet alone [27]. Metformin could improve immediate pregnancy outcomes, reducing excess fetal growth, adiposity, and pregnancy-related hypertensive disorders [28]. However, the role of metformin is still controversial [29,30]. In Vietnam, this medicament is not yet recommended to treat diabetes in pregnancy according to the Minister of Health. Recently, Simmons et al. have shown that immediate treatment of gestational diabetes before 20 weeks’ gestation led to a modestly lower incidence of a composite of adverse neonatal outcomes than no immediate treatment [31]. Early diagnosis along with timely management helps in reducing adverse materno–fetal outcomes during pregnancy [32].

Currently, some evidence has demonstrated that medical nutrition therapy plays a potential role in decreasing dramatically the adverse effects of GDM [33,34]. Moreover, aside from the importance of diet, physical activities with moderate exercise and frequent self-monitoring became a cornerstone of recommendations in preventing the severity of GDM [35,36]. In 2021, Yaping et al. conducted a randomized controlled trial including 50 cases in the control group and 51 cases in the experimental group. The study found that moderate-intensity aerobic exercise can help improve blood glucose control and insulin use in patients with GDM [37].

Nevertheless, there was a paucity of studies examining precision lifestyle-based interventions for GDM [38,39]. The knowledge of nutrition and normoglycemia control among Vietnamese pregnant women remains limited [21,40]. There also has been modest research gaps on the association between an appropriate intervention and an initially elevated fasting glycemic parameter. Therefore, this study aims to assess the association between dietary–physical management and the results of glycemic control after identification of the positive fasting glycemic index in OGTT. Additionally, the secondary purpose was to reveal some relative factors to successful control of glycemia.

## 2. Materials and Methods

### 2.1. Study Design, Setting, and Population

This retrospective cross-sectional study was carried out between 4/2022 and 7/2022 at Hung Vuong Hospital, which is a 900-bed tertiary care and referral hospital located in the south of Vietnam (Figure 1). The study enrolled all the pregnant women who had been already diagnosed gestational diabetes mellitus (GDM) using the 75-g oral glucose tolerance test (OGTT) 2 weeks earlier. In addition, the participants presented at least the occasion of the high fasting glycemic parameter ≥ 92 mg/dL (≥5.1 mmol/L). The one-step screening method was applied according to criteria of the International Association of Diabetes and Pregnancy Study Groups (IADPSG) and the World Health Organization (WHO) [12,41]. Glucose levels were collected from antecubital venous blood samples just before and at the first and second hour after taking 200 mL oral glucose solution containing 75 g glucose following 8 h of fasting in the morning. The blood-containing tube was sent to the laboratory within 30 min after blood sample collection. The blood glucose test was measured using an automatic analysis machine (Anility, Abbott, Chicago, IL, USA). The investigators were trained during this study period.

### 2.2. Sample Size

The study sample size was calculated based on a formula using estimated prevalence for cross-sectional study as follows: $n=\frac{Z^{2}P\left(1-P\right)}{d^{2}}$ [42], using *Z* = 1.96 of 95% confidence level, precision d of 0.05, and the expected proportion *P* of 0.872 for the GDM pregnant women reaching the controlled target after 7 days of dietary–physical management at Hung Vuong Hospital [43]. The study collected non-probabilistic samples and required a minimum of 172 GDM pregnant women presenting at the out-patient antenatal care unit, Hung Vuong Hospital. The participants had at least a high fasting glycemic parameter at OGTT and were educated with about dietary–physical plan. During 3 months, the study enrolled finally 174 Vietnamese singleton pregnant women who met inclusion criteria after excluding 15 participants not satisfying the inclusion criteria.

### 2.3. Inclusion and Exclusion Criteria

Singleton pregnancies with gestational diabetes mellitus were included with the accurate gestational age based on the first-trimester ultrasound and the date of the last period. The participants recorded an elevated index of fasting glucose concentration and the HbA1C level was normal. The patients had no relevant history of metabolic diseases. In addition, the patients had no mental disorders and could complete the self-administered Pregnancy Physical Activity Questionnaires (PPAQs) using the Vietnamese version.

Exclusion criteria included abnormal pregnancies such as congenital defects, placenta previa, fetal growth restriction, intra-uterine fetal death, and fetal distress. Also excluded were patients who did not follow-up strictly the visit on the scheduled date (earlier than 2 weeks or later than 3 weeks after the first diagnosis of GDM), participants who had diabetes before pregnancy and were treated with insulin therapy, patients who had a medical condition that is contraindicated to practice exercises, and participants who refused to receive the dietary plan and physical compliances following the hospital protocol. The patients who were required to be hospitalized due to other medical indications were also excluded, as were deaf and dumb patients who could not communicate with the investigators, and those who could not read and complete the Vietnamese items. Finally, participants receiving the corticosteroid therapy for antenatal lung maturation (dexamethasone, betamethasone) within approximately 72 h prior to the blood glucose test were also excluded due to the risk of hyperglycemia [44,45].

### 2.4. Dietary and Physical Intervention

The patients were managed with an adequate intake of nutrients with a low-carbohydrate diet (<35–45% of daily caloric intake from carbohydrates) following the practical protocol of Hung Vuong Hospital and international evidence-based guidelines during antenatal visits [23].

Nutrition Counselling: Three main meals including breakfast, lunch, and dinner accompanied by 2–3 snacks were required. Skipping meals was not recommended. The pregnant women were advised to use a dish with a 20 cm diameter which was divided into 3 parts (Figure 2). One quarter part consists of protein (fish, carb, egg, beef, chicken, cheese, unsweetened cow’s milk, etc.). The other quarter part consists of carbohydrate-containing foods (rice, bread, noodle, sticky rice, rice porridge, sweet potatoes, popcorn, etc.), and the half part includes vegetables and fruits. The vegetables could be cooked with oil or could be eaten fresh. Some other products such as yogurt and all kinds of beans could also supply carbohydrates. Saturated fat and cholesterol must be reduced in a meal.

Physical advice: The pregnant women were advised to practice yoga, jogging, swimming, and exercises. The preconception was that physical activities should be continued during pregnancy. The exercises may increase from light intensity to moderate intensity. The time duration of physical activity should last at least 30 min per day. Jogging or exercises with hands for 10 min were recommended after meals. Pregnant women should intake enough water and maintain their exercises regularly in pregnancy.

No pharmacological method was used in this study for the treatment of GDM women. Meanwhile, the participants could use supplementation including iron-folic acid, zinc, calcium, vitamin D3, vitamin B12, and multiple micronutrient intake.

The retrospective evaluation of glycemic concentration was performed after 2 weeks of routine antenatal management following the questionnaire relating to physical activities of Chasan–Taber et al. which was validated following the Vietnamese version of the pregnancy physical activity questionnaires (PPAQs) of Ota et al. [46,47].

### 2.5. Outcomes

The data were recorded directly by the study protocol and collected through the out-patient file. Physical activities were measured following the metabolic equivalent of energy (MET) (MET hours per week) based on self-evaluation of PPAQs. The PPAQ is a semiquantitative questionnaire that asks respondents to report the time spent participating in 32 activities including household/caregiving (13 activities), occupational (5 activities), sports/exercise (8 activities), transportation (3 activities), and inactivity (3 activities) [46]. Each activities were classified into 4 groups including sitting activity <1.5 METs, light-intensity activity: 1.5– < 3.0 METs, moderate-intensity activity: 3.0– < 6.0 METs, and vigorous-intensity activity: ≥6.0 METs [47].

The pre-pregnancy body mass index (kg/m^2^) was classified following the Asian–Pacific guideline [48]. Overweight gain during pregnancy was determined according to Institute of Medicine/National Research Council guidelines [49].

The optimal glycemic target of GDM was examined after 2 weeks of outpatient management. The participants were given instructions to fast from midnight and present in the morning for testing. At admission, the blood glucose concentration of the participants was checked with venous blood samples for the measurement of fasting blood glucose concentration after at least 8 h without a meal and the blood glucose sample at 2 h post-meals. The glycemic goal was considered as successful achievement/optimal glycemic control if both fasting and 2 h postprandial parameters of blood glucose were recorded below 95 mg/dL (5.3 mmol/L) and 120 mg/dL (6.7 mmol/L), respectively [50,51]. The failed group or suboptimal glycemic control was defined as ≥1 occasion of either fasting glucose ≥ 95 mg/dL or 2 h postprandial glucose ≥ 120 mg/dL. The glycemic value (mg/dL) was rounded to the second decimal place.

### 2.6. Statistical Analysis

The data were entered using Microsoft Office Excel software version 14.0 and analyzed by using the Statistical Package for the Social Sciences (SPSS) 20.0 (Armonk, NY, USA) and generated graphs by using SPSS 26.0 (Armonk, NY, USA). Descriptive statistics were expressed as means and standard deviations (X ± SD), and median and interquartile range for quantitative variables depending on the distribution of data. Frequency data comparisons were performed across categories using the χ^2^ (chi-square test) and Fisher-exact test as appropriate. Pearson’s correlation (depending on the parametric distribution of data) was used to determine the correlation coefficient (*r*) between two variables, which ranged from −1 to +1, indicating a negative and positive correlation, respectively. An |*r*| value ≥ 0.8 indicated a very strong correlation, |*r*| = 0.6–0.7 indicated a moderate correlation, 0.2 < |*r*| < 0.6 indicated a fair correlation, and *r* ≤ 0.2 indicated a poor correlation [52]. The odds ratio (OR) and adjusted odds ratios (aORs) with 95% confidence intervals (CI) were calculated according to univariate and multivariable logistic regression. Multivariable logistic regression was calculated based on all variables with *p* < 0.25 from univariate logistic regression and adjusted for maternal age, demographic region, parity, pre-pregnancy BMI, overweight gain, OGTT results, moderate intensive activities, working activities, exercises/sports activities, and total intensity of physical activities. Statistical significance was defined as *p*-value < 0.05 for all analyses.

### 2.7. Ethics Statement

Informed consent was obtained from all the participants. The study was approved by the Ethics committee of Pham Ngoc Thach University of Medicine with the approval of 582/TĐHYKPNT-HĐĐĐ on 25 January 2022.

## 3. Results

During 3 months, the study recruited 174 women who met the adequate inclusion criteria. The mean age of the participants was 31.95 ± 5.45 years old and almost all of the cases were in the age group < 35 years old (67.2%). More than half of cases had ≥ one birth and BMI ≥ 25 kg/m^2^, 60.3%, and 58.6%, respectively. The success rate of GDM control after 2 weeks of GDM management was 59.2% (Table 1).

In this study, the mean glycemic value of fasting and 2 h postprandial parameters after 2 weeks of GDM management were significantly lower in the successful control group compared with that of the failed control group (Table 2 and Figure 3 and Figure 4). In addition, the success rate of glycemic control in the group with a high fasting index of glycemic value was 69.6% 95% CI (62.7–76.4) and decreased gradually in the groups with two and three indexes of OGGT, 56.5%, and 49.2%, respectively (Table 3).

Table 4 shows the correlation between the test of blood glucose at OGTT and after 2 weeks of management. The study found a moderate-weak correlation among the blood glucose level parameters at OGTT as well as between fasting and 2 h-postprandial blood glucose concentrations after 2 weeks of management. However, the study revealed only a fair correlation between the corresponding test of blood glucose at OGTT and after 2 weeks of management (Figure 5).

In the overall study, almost all participants experienced the sitting activity and light-intensity activity compared to moderate- and vigorous-intensity activities. The Vietnamese pregnant women spent the most time on household/caregiving activities and occupational/working activities (51.36 ± 33.29 and 44.42 ± 49.63) (Met-hr/wk). The study found a poor correlation of physical exercise/sports activities between the failed and successful glycemic control groups since physical exercise/sports only occupied approximately 1.51 ± 1.50 (Met-hr/wk) in our study population (Table 5 and Appendix A).

Using univariate regression, our study found some factors related to the result of glycemic control including maternal age, BMI, overweight gain, positive OGTT result with three high indexes, and exercise/sport activities. Using multivariable analysis, after adjustment for confounding variables, the study revealed the most important factor affecting the success rate of glycemic control was the initial OGTT results. Compared to the separate fasting glycemic parameter group, the group of two and three glycemic parameters reduced significantly the success rate of 69.0% (OR: 0.31, 95% CI (0.12–0.76), *p* < 0.01) and 63.0% (OR: 0.37, 95% CI: 0.16–0.81, *p* < 0.02) (Table 6).

## 4. Discussion

In the present study, the one-step screening (75-g OGTT) was used since this approach is recommended widely by the International Association of Diabetes and Pregnancy Study Groups (IADPSG) and the World Health Organization (WHO). In addition, the one-step screening is currently adapted at our center with cost considerations, patient preference, and availability of infrastructure locally [53]. In the United States, the common approach to detecting gestational diabetes mellitus (GDM) is the two-step protocol recommended by the American College of Obstetricians and Gynecologists (ACOG) [54]. There were no significant differences between the one-step and two-step group in the risks of perinatal and maternal complications [5].

Along with the appropriate detection of GDM, early management applying diet therapy combined with physical activity could reduce substantially the adverse materno–fetal outcomes [55]. The findings of our study showed that the number of GDM women with a separate fasting glycemic parameter reached the controlled target better than the contributory group with two or three high glycemic parameters. Several studies have demonstrated that a high postprandial glycemic threshold is commonly associated with more severe outcomes. A recent study has shown the rates of postpartum type 2 diabetes mellitus (T2DM) and prediabetes were significantly higher in the suboptimal glycemic control group than in the optimal glycemic control group: 22.4% vs. 3.0%, *p* < 0.001 for T2DM and 45.3% vs. 23.5%, *p* < 0.001 for prediabetes [16]. In multivariable log-binomial regression, Schwartz et al. found that the mean postprandial glucose was significantly associated with GDM recurrence (*p* = 0.017) after adjusting for maternal age, family history of diabetes, insulin use, and inter-pregnancy interval. Thus, the study concluded that tighter postprandial glycemic control should be considered [56].

Meanwhile, according to Hofer et al., the use of tighter glycemic targets in women with GDM does not change the concentrations of maternal and infant biomarkers compared to less tight targets. However, when compliance is achieved to tighter targets, maternal and infant biomarkers are altered [57]. A secondary analysis of this study also revealed that the tighter glycemic treatment targets and the less tight one in GDM women did not impact their mental health status at 36 weeks of gestation and at 6 months postpartum [22]. Therefore, standard monitoring of blood glucose in uncomplicated GDM women may be sufficient in low-resource settings and LMICs. Instead, among high-risk pregnant women such as overweight and obese women, intensive glycemic targets can be used, and this approach results in improved glycemic control when compared to standard glycemic targets [58]. In addition, real-time continuous glucose monitoring can be useful in women who received insulin therapy to avoid hypoglycemic risk where applicable [59]. Our study also revealed a fair correlation between the fasting blood glucose level and 2 h blood glucose at OGTT and after 2 weeks of management, respectively. This finding demonstrated the effectiveness of treatment with medical nutrition therapy and a physical activity program. The initially elevated blood glucose levels could be controlled with adequate management. According to the community-based study implementing twenty-seven studies with 6242 GDM women, diet therapy was more effective than usual care in improving postprandial blood glucose levels [60].

Similar to the report of Ota et al., almost all Vietnamese pregnant women spent time doing sedentary/light and moderate intensity activity through household/caregiving activities (38.0 Met-hr/wk) [47]. Using univariate regression analysis, our study found that maternal age, parity, body mass index (BMI), overweight gain during pregnancy, OGTT results, and exercise/sport activities were related to the success rate of glycemic control. Body mass index in early pregnancy was known as a risk factor for GDM; thus, the optimal BMI in early pregnancy is the key to preventing GDM. Any efforts aim to control maternal weight resulting in preventing GDM [61]. In our study, using multivariable regression analysis, there was only one factor impacting the result of glycemic control, which was the initial OGTT results. Furthermore, in accordance with Morikawa et al., among 72 pregnant women with GDM during the first pregnancy, 21 pregnant women scored two or three positive points on a 75-g OGTT during their first pregnancies; the GDM recurrence rate among these women (66.7%) was significantly higher than that among the 51 pregnant women who scored one positive point (39.2%; *p* = 0.0411) [62].

Based on eleven studies in an literature review, Onaade et al. recommended that increased non-supervised physical activity and lifestyle behavioral change be implemented for successful glycemic control of pregnant and postpartum populations [36]. GDM causes a burden on socio-economic status, whereas a large proportion of GDM risk can be preventable with lifestyle modification [1,63]. According to the findings of Yaping et al., the mean fasting glycemic level in the experimental group with moderate aerobic exercises was lower than in the control group, 4.92 ± 0.15 vs. 5.08 ± 0.17 (mmol/L), *p* < 0.0001, respectively. Similarly, the result was statistically significant when evaluating the value of the mean 2 h postprandial blood glucose level, 6.11 ± 0.11 vs. 6.25 ± 0.22 (mmol/L) [37]. In line with Nwamaka Igwesi-Chidobe et al., postprandial blood glucose levels were better improved by regular supervised exercise plus daily brisk walks than routine obstetric care or no treatment. Diet and exercise were superior to diet alone in reducing maternal weight gain [60].

### 4.1. Strengths and Limitations

The study was conducted at the consultant hospital in the south of Vietnam and was conducted with a strict protocol. Furthermore, the study design was retrospective cross-sectional analysis; thus, the findings of the natural study reflect optimally the GDM management of the hospital, without requiring further intervention in the present study. Additionally, the study used the PPAQs that were translated into a Vietnamese version, and this questionnaire tool has been validated in the previous survey.

However, the study was carried out in a short period of time with a small number of Vietnamese participants. The non-random sampling study could not be representative for the GDM pregnant women. Moreover, our study did not recruit GDM pregnant women without the high fasting glycemic parameter and evaluate its success rate and reveal its related factors. To reduce the bias, the investigators were well trained, and the pregnant women received explanations in detail during the survey. In addition, the blood glucose sample was withdrawn carefully and was sent to the laboratory as rapidly as possible (less than 30 min). Furthermore, the glycemia result was analyzed using a limited number of analyzers.

### 4.2. Future Implications

A multicentric study with a large number of participants including the control group without a high fasting glycemic index is required to assess the success rate and its related factors. Furthermore, the participants could wear a digital pedometer to monitor accurately the physical activities and to count the walking steps during daily activities in more than 2 weeks (4–6 weeks). The optimal need for dietary modification and energy for exercises can be calculated individually based on these initial assessments in preventive strategies.

## 5. Conclusions

The success rate of blood glucose control was highest in the group with a separately high fasting parameter and lower in the groups with two and three high parameters. Therefore, GDM pregnant women with two or three high glycemic parameters should be strictly managed after diagnosis in antenatal care management. Additionally, pregnant women participating in high-intensity sports are more likely to increase their success rate compared with pregnant women practicing light and moderate-intensity physical activities. Future research is required to strengthen the findings of these aspects in healthcare policy.

## Figures and Tables

**Figure 1 jcm-13-05895-f001:**
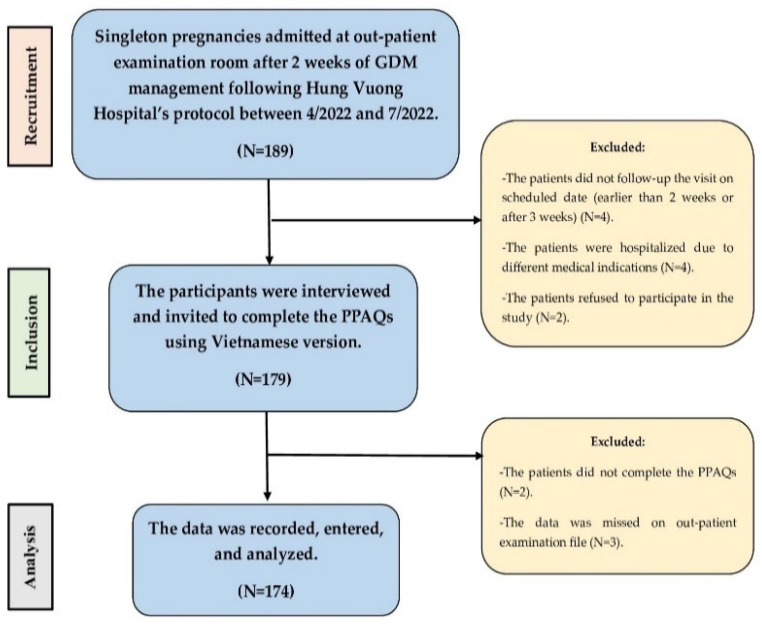
Study flowchart.

**Figure 2 jcm-13-05895-f002:**
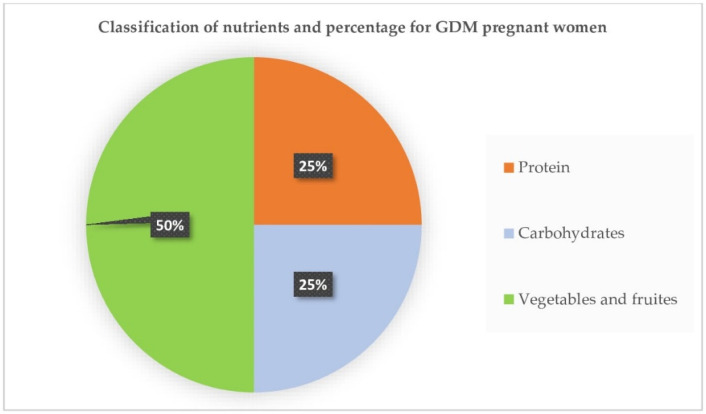
Three categories of nutrients and corresponding ratio for GDM pregnant women in the present study.

**Figure 3 jcm-13-05895-f003:**
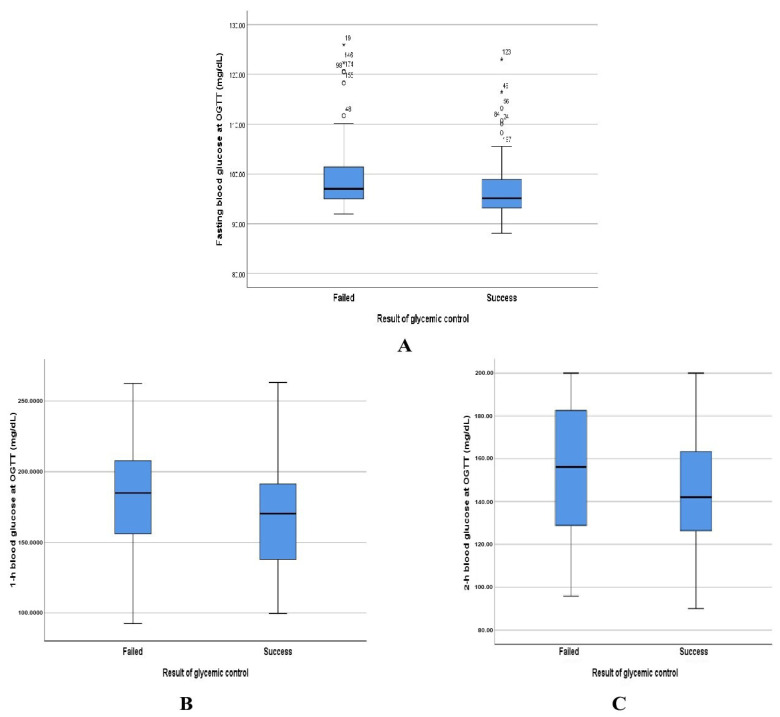
Box plots show the difference in blood glucose concentration between the failed and success groups at initial OGTT results: (**A**) fasting blood glucose concentration (**B**) 1 h blood glucose concentration (**C**) 2 h blood glucose concentration, respectively. *: Outlier value.

**Figure 4 jcm-13-05895-f004:**
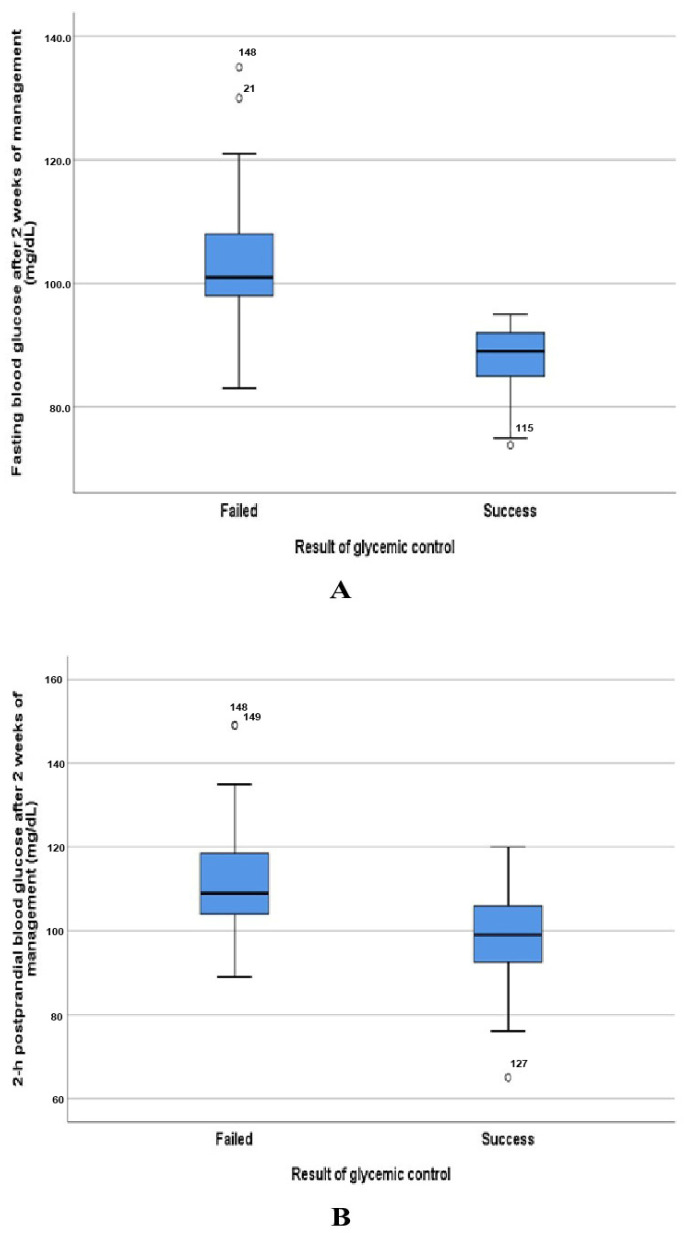
Box plots show the difference in blood glucose concentration between the failed and success groups after 2 weeks of GDM management: (**A**) fasting blood glucose concentration (**B**) 2 h postprandial blood glucose concentration, respectively.

**Figure 5 jcm-13-05895-f005:**
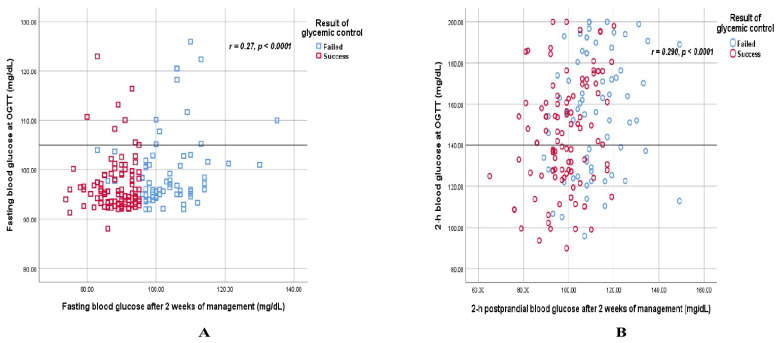
Linear regression graphs display a weak correlation between fasting blood glucose at OGTT and after 2 weeks of GDM management (**A**) and 2 h blood glucose at OGTT and after 2 weeks of GDM management (**B**).

**Table 1 jcm-13-05895-t001:** Baseline characteristics of the study population.

Characteristics		Frequency (n = 174)	Percentage (%)
Maternal age (years)	mean ± SD (min-max)	31.95 ± 5.45 (20–47)	
	<35	117	67.2
	≥35	57	32.8
Gestational age (weeks)	mean ± SD	26.55 ± 1.70	
Demographic region	Urban	120	69.0
	Rural	54	31.0
Occupation	Farmer/worker/ seller/housework	163	93.7
	Office staff/student	11	6.3
Education	Primary school	6	3.4
	Secondary school	48	27.6
	High school	71	40.8
	College/university	49	28.3
Average monthly income (million VNĐ per month)	<10	101	58.0
	10–20	68	39.1
	>20	5	2.9
Familial history of diabetes (in any first-degree relatives)	Yes	30	17.2
	No	144	82.8
Diabetes mellitus in the previous pregnancies	Yes	16	9.2
	No	158	90.8
Parity (times)	0	69	39.7
	≥1	105	60.3
BMI (kg/m^2^)	mean ± SD (min-max)	26.06 ± 3.56 (18.73–36.89)	
	Underweight (<18.5)	0	0.0
	Normal weight (18.5–22.9)	31	17.8
	Overweight (23.0–24.9)	41	23.6
	Obese (≥25.0)	102	58.6
Overweight gain (kg)	Yes	49	28.2
	No	125	71.8
OGTT result	One high parameter	69	39.7
	Two high parameters	46	26.4
	Three high parameters	59	33.9
Glycemic value (mg/dL)	Fasting parameter at OGTT mean ± SD (min-max)	97.88 ± 6.51 (88.07–126.00)	
	1-h parameter at OGTT mean ± SD (min-max)	173.00 ± 37.16 (92.33–263.32)	
	2-h parameter at OGTT mean ± SD (min-max)	149.39 ± 28.16 (90.00–200.00)	
Physical activities (Met-hr/wk)	Daily activities (mean ± SD)	73.48 ± 36.33	
	Doing exercises/ sports (mean ± SD)	56.45 ± 52.19	
Glycemic value after 2 weeks of GDM management (mg/dL)	Fasting glycemic parameter (mean ± SD) (min-max)	94.37 ± 10.06 (73.80–135.00)	
	2-h postprandial glycemic parameter (mean ± SD) (min-max)	103.61 ± 13.35 (65.00–149.00)	
Result of GDM control after dietary–physical management	Successful control	103	59.2
	Failed control	71	40.8

All data are presented as mean ± SD or n (%).

**Table 2 jcm-13-05895-t002:** The value of glycemic parameters at OGTT and after 2 weeks of GDM management.

	Glycemic Control Group	Success Group (N = 103)	Failed Group (N = 71)	*p*-Value *
Glycemic Parameters (mg/dL)				
Fasting blood glucose	At OGTT	96.72 ± 5.42	99.56 ± 7.55	0.007
	After 2 weeks of GDM management	88.23 ± 5.04	103.27 ± 8.79	<0.001
2 h blood glucose	At OGTT	144.70 ± 26.90	156.19 ± 28.73	0.009
	After 2 weeks of GDM management	98.46 ± 11.03	111.10 ± 12.92	<0.001

* Sig (2-tailed) from Independent sample test value.

**Table 3 jcm-13-05895-t003:** The success rate of glycemic control according to the initial OGTT results.

Glycemic Parameters of OGTT Results	Frequency (n)	Success Rate of Glycemic Control n (%)	95% CI (Lower-Upper Limit)	*p*-Value
One high parameter (fasting blood glucose level)	69	48 (69.6)	62.7–76.4	0.059
Two high parameters (fasting and 1-h or 2-h blood glucose levels)	46	26 (56.5)	49.1–63.8	
Three high parameters (fasting, 1-h, and 2-h blood glucose levels)	59	29 (49.2)	41.7–56.6	
Total	174	103 (59.2)	51.9–66.5	

**Table 4 jcm-13-05895-t004:** Pearson correlation between test of blood glucose at OGTT and after 2 weeks of GDM management.

Test of Blood Glucose at OGTT and after 2 Weeks of GDM Management	Fasting Blood Glucose at OGTT (mg/dL)	1 h Blood Glucose at OGTT (mg/dL)	2 h Blood Glucose at OGTT (mg/dL)	Fasting Blood Glucose after 2 Weeks of GDM Management (mg/dL)	2 h Postprandial after 2 Weeks of GDM Management (mg/dL)
Fasting blood glucose at OGTT (mg/dL)	-	*r* = 0.341 ** *p* < 0.0001	*r* = 0.302 ** *p* < 0.0001	*r* = 0.270 ** *p* < 0.0001	*r* = 0.264 ** *p* < 0.0001
1 h blood glucose at OGTT (mg/dL)	*r* = 0.341 ** *p* < 0.0001	-	*r* = 0.699 ** *p* < 0.0001	*r* = 0.129 *p* = 0.090	*r* = 0.292 ** *p* < 0.0001
2 h blood glucose at OGTT (mg/dL)	*r* = 0.302 ** *p* < 0.0001	*r* = 0.699 ** *p* < 0.0001	-	*r* = 0.167 * *p* = 0.027	*r* = 0.290 ** *p* < 0.0001
Fasting blood glucose after 2 weeks of GDM management (mg/dL)	*r* = 0.270 ** *p* < 0.0001	*r* = 0.129 *p* = 0.090	*r* = 0.167 * *p* = 0.027	-	*r* = 0.459 ** *p* < 0.0001
2 h postprandial after 2 weeks of GDM management (mg/dL)	*r* = 0.264 ** *p* < 0.0001	*r* = 0.290 ** *p* < 0.0001	*r* = 0.290 ** *p* < 0.0001	*r* = 0.459 ** *p* < 0.0001	-

* Correlation is significant at the 0.05 level (2-tailed), ** Correlation is significant at the 0.01 level (2-tailed).

**Table 5 jcm-13-05895-t005:** Features of physical activities following intensity and types.

Activities	Frequency (n)	Percentage (%)
Sitting/sedentary (Met-hr/wk)	47.63 ± 25.06 *	
Low	84	48,3
High	90	51.7
Light intensity (Met-hr/wk)	62.25 ± 46.71 *	
Low	87	50.0
High	87	50.0
Moderate intensity (Met-hr/wk)	8.67 (0.00–126.80) ** (5.25–21.67) ***	
Low	82	47.1
High	92	52.9
Vigorous intensity (Met-hr/wk)	0.00 (0.00–84.0) ** (0.00–0.00) ***	
Low	163	93.7
High	11	6.3
Total intensity of activities (Met-hr/wk)	129.94 ± 67.86 *	
Low	85	48.9
High	89	51.1
Household/caregiving activity (Met-hr/wk)	51.36 ± 33.29 *	
Low	79	45.4
High	95	54.6
Occupational/working activity (Met-hr/wk)	44.42 ± 49.63 *	
Low	85	48.9
High	89	51.1
Physical exercises/sports (Met-hr/wk)	1.51 ± 1.50 *	
Low	63	36.2
High	111	63.8

* Mean ± standard deviation, ** min-max, *** IQR (Q1–Q3); Physical activity converted by median including 2 values:—Low: The intensity of activity has a value smaller than the median of the intensity of activity in each activity level (sitting, light, moderate, vigorous), and each type of activity (household, working, sports).—High: The intensity of activity has a value greater than or equal to the median of the intensity of activity at each level of movement (sitting, light, moderate, strong), and each type of movement (household, working, sports).

**Table 6 jcm-13-05895-t006:** Factors related to the success rate of glycemic control.

Factors	Classified	Success (n = 103) n (%)	Failed (n = 71) n (%)	OR * (95% CI)	*p*-Value *	aOR ** (95% CI)	*p*-Value **
Maternal age (years)	<35	77 (65.8)	40 (34.2)	Ref	-	Ref	
	≥35	26 (45.6)	31 (54.4)	0.44 (0.23–0.83)	0.01	0.65 (0.30–1.44)	0.29
Geographic region	Urban	67 (55.8)	53 (44.2)	Ref	-	Ref	
	Rural	36 (66.7)	18 (33.3)	1.58 (0.81–3.10)	0.18	1.97 (0.91–4.26)	0.09
Occupation	Worker	8 (72.7)	3 (27.3)	Ref	0.35	NA	
	Office staff	95 (58.3)	68 (41.7)	0.52 (0.13–2.05)			
Education	Primary school	4 (66.7)	2 (33.3)	Ref	-	NA	
	Secondary school	30 (62.5)	18 (37.5)	0.83 (0.14–5.03)	0.84		
	High school	36 (50.7)	35 (49.3)	0.51 (0.09–2.99)	0.46		
	College/ university	33 (67.3)	16 (32.7)	1.03 (0.17–6.25)	0.97		
Average monthly income (million VNĐ per month)	<10	56 (55.4)	45 (44.6)	Ref	-	NA	
	10–20	44 (64.7)	24 (35.3)	1.47 (0.78–2.78)	0.23		
	>20	3 (60.0)	2 (40.0)	1.20 (0.19–7.52)	0.83		
Parity	0	49 (71.0)	20 (29.0)	Ref	-	Ref	
	≥1	54 (51.4)	51 (48.6)	0.43 (0.23–0.83)	0.01	0.52 (0.23–1.17)	0.11
Pregestational BMI (kg/m^2^)	Normal (18.5–22.9)	23 (74.2)	8 (25.8)	Ref	-	Ref	
	Overweight (23.24.9)	26 (63.4)	15 (36.6)	0.60 (0.22–1.68)	0.33	0.50 (0.16–1.62)	0.25
	Obese (≥25.0)	54 (52.9)	48 (47.1)	0.39 (0.16–0.96)	0.04	0.55 (0.19–1.55)	0.26
Overweight gain during pregnancy	Yes	22 (44.9)	27 (55.1)	Ref	-	Ref	
	No	81 (64.8)	44 (35.2)	2.26 (1.15–4.42)	0.02	1.71 (0.78–3.76)	0.18
Initial result of OGTT (mmol/L)	One elevated parameter	48 (69.6)	21 (30.4)	Ref	-	Ref	
	Two elevated parameters	26 (56.5)	20 (43.5)	0.57 (0.26–1.24)	0.15	0.31 (0.12-0.76)	0.01
	Three elevated parameters	29 (49.2)	30 (50.8)	0.42 (0.21–0.42)	0.02	0.37 (0.16–0.86)	0.02
Moderate intensive activities	Low	44 (53.7)	38 (46.3)	Ref	-	Ref	
	High	59 (64.1)	33 (35.9)	1.54 (0.84–2.83)	0.16	1.17 (0.51–2.66)	0.71
Working activities	Low	46 (54.1)	39 (45.9)	Ref	-	Ref	
	High	57 (64.0)	32 (36.0)	1.51 (0.82–2.77)	0.18	0.60 (0.22–1.61)	0.31
Exercises/sports activities	Low	30 (47.6)	33 (52.4)	Ref	-	Ref	
	High	73 (65.8)	38 (34.2)	2.11 (1.12–3.97)	0.02	1.83 (0.82–4.10)	0.14
Total intensity of physical activities	Low	45 (52.9)	40 (47.1)	Ref	-	Ref	
	High	58 (65.2)	31 (34.8)	1.66 (0.90–3.06)	0.10	1.87 (0.69–5.13)	0.22

* Univariate regression, ** Multivariable regression (including all variables with *p* < 0.25). The data are presented as adjusted odds ratios (aORs) with 95% CI. NA: not applicable.

## Data Availability

The data used to support the findings of this study are available from the corresponding author upon request. Data availability is restricted due to legal policy of our hospital and ethical reasons.

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
