# Peer review of "Non-Pharmacological Management of Gestational Diabetes Mellitus with a High Fasting Glycemic Parameter: A Hospital-Based Study in Vietnam"

_jcm, 2024, doi:10.3390/jcm13195895_

Round 1
Reviewer 1 Report
Comments and Suggestions for Authors
Overall Comments:
The authors present a retrospective cross-sectional study on 174 Vietnamese pregnant women with the positive oral glucose tolerance test examining the success rate of Gestational Diabetes Mellitus (GDM) with high parameters of fasting blood glucose concentration which was managed after 2 weeks of a dietary regimen combined with adequate exercise and to reveal its related factors.
The authors found that the application of diet and physical activity among GDM pregnant women is potentially necessary to improve the effectiveness of treatment, minimize adverse pregnancy outcomes, and reduces dramatically the hospitalization rate.
I found the study well designed and developed, the question posed by the authors is well defined in page 2 (lines 62-64): “However, 70% of GDM could be successfully achieved by adequate dietary plans and physical activities. Preventing excessive gestational weight gain is a necessary treatment in the underlying management of GDM.”
The methods are appropriate and very interesting from the methodological-scientific point of view, an intensive data collection work has been done by authors.
The discussion and conclusions are well-balanced and adequately supported by the data.
I found that from the research point of view, the study has merit however a minor correction is necessary.
Comment:
Page 3: The Flow Chart Diagram contains scarce and confusing information. It should be elaborated following a model that provides more information (participants included and excluded) about each phase of the study, such as Enrollment/Allocation/Follow-up/Analysis,
Author Response
Dear reviewer 1,
We appreciate the time and effort that the reviewers dedicated to providing feedback on our manuscript and are grateful for the insightful comments and valuable improvements to our manuscript. In response to your comments, we have carefully revised the manuscript, and we incorporated the suggestions made by the reviewers. We indicated them by using tracked changes in revised paper.
Reviewer Comments:
Reviewer 1:
Page 3: The Flow Chart Diagram contains scarce and confusing information. It should be elaborated following a model that provides more information (participants included and excluded) about each phase of the study, such as Enrollment/Allocation/Follow-up/Analysis.
Authors response: Thank you for your positive comments and your revision.
We have amended it and modified the study flowchart following your great suggestion.
In summary, we have made substantial revisions to the manuscript to ensure that the points raised by the reviewers have been addressed thoroughly. We did not list all the changes in this letter but marked using tract change in revised paper.
We believe that these revisions have significantly improved the manuscript and hope that the reviewer and the editorial board will find the updated version satisfactory.
Once again, we would like to take this opportunity to thank you again for all your time involved and we hope the revised manuscript could be acceptable for you. We hope the revised manuscript and all the detailed responses will clarify for all your queries. We are still waiting for your decision and further instructions.
On behalf of authors,
Corresponding author,
Phuc Nhon Nguyen MD, MSc
Tu Du Hospital, Ho Chi Minh City, Vietnam
Reviewer 2 Report
Comments and Suggestions for Authors
jcm-3082837
It is a study that evaluated the effect of a dietary regimen combined with exercise on fasting glucose concentration two weeks after the start of the non-pharmacological regimen.
The authors must clearly specify what the dietary regimen consisted of and how adherence to the regimen was evaluated. Additionally, it should be described whether the physical exercise was a program or if only the usual physical activity performed by the participants was assessed.
Another issue that the authors need to clarify is whether any sampling was done and what type it was. What diagnostic criteria were used?
Were only pregnant women with gestational diabetes included, or were pregnant women with carbohydrate intolerance also included?
Therefore, it is necessary for the authors to clarify all doubts in order to consider the validity of the study.
The title is too long and does not reflect the content of the study.
Introduction
The context, rationale, and objective of the study are appropriate.
Citations should be presented in a single pair of brackets or parentheses when there is more than one, for example [1,2]
It is important for the authors to mention what causes the wide variation in the diagnosis of diabetes according to the criteria.
The sentence on lines 60 and 61, what is its relationship to gestational diabetes?
Materials and methods
In the materials and methods section, the authors should specify the following points:
What was the pharmacological treatment of the participants?
In which cases was the pharmacological treatment used?
What did the pharmacological treatment consist of?
Are the authors sure that their methodological design is descriptive and retrospective?
What criteria did you use to diagnose gestational diabetes?
Were all the participants diagnosed with gestational diabetes, or were pregnant women with carbohydrate intolerance also included?
What type of sampling was conducted for the selection of the 174 participants?
What does PPAQ mean? Whenever an abbreviation is used for the first time, it must be defined.
What did the dietary and physical intervention consist of? Both interventions need to be described.
Adherence to the interventions was evaluated.
What criteria were used to evaluate the dietary and physical intervention at 2 weeks? It's not a very short period.
Physical activity questionnaires during pregnancy are validated for the group of Vietnamese pregnant women.
It is not clear whether the authors only assessed the participants' usual physical activity or evaluated a physical activity program.
If the authors only assessed the participants' usual physical activity, physical activity should not be included in the title, as it gives the impression that the participants underwent a physical activity program along with a medical nutrition program.
Authors must specify all measures taken to address potential sources of bias.
Results
In the results section, the authors need to improve Figure 2.

Author Response
Dear reviewer 2,
We appreciate the time and effort that the reviewers dedicated to providing feedback on our manuscript and are grateful for the insightful comments and valuable improvements to our manuscript. In response to your comments, we have carefully revised the manuscript, and we incorporated the suggestions made by the reviewers. We indicated them by using tracked changes in revised paper.
Reviewer 2:
-The title is too long and does not reflect the content of the study.
Authors response: Thank you for your correction. We changed the shorter title: “Non-pharmacological management of gestational diabetes mellitus with a high fasting glycemia parameter: a hospital-based study in Vietnam”
Introduction
-The context, rationale, and objective of the study are appropriate.
Authors response: Thank you for your positive comment.
-Citations should be presented in a single pair of brackets or parentheses when there is more than one, for example [1,2].
Authors response: We corrected this point and incorporate in overall paper.
-It is important for the authors to mention what causes the wide variation in the diagnosis of diabetes according to the criteria.
Authors response: The choice of criterion with different measurements can affect diabetes prevalence and the diabetes could be underestimated and underdiagnosed depending on the resource settings. (NCD Risk Factor Collaboration (NCD-RisC). Global variation in diabetes diagnosis and prevalence based on fasting glucose and hemoglobin A1c. Nat Med. 2023;29(11):2885-2901. doi:10.1038/s41591-023-02610-2). In this report, concerning fasting plasma glucose and hemoglobin A1C, the 117 population-based study concluded that: “In most low- and middle-income regions, isolated elevated HbA1c was more common than isolated elevated FPG. In these regions, the use of FPG alone may delay diabetes diagnosis and underestimate diabetes prevalence. Our prediction equations help allocate finite resources for measuring HbA1c to reduce the global shortfall in diabetes diagnosis and surveillance.”
Similarly, in the report of Nguyen et al., the prevalence of GDM varied considerably by the diagnostic criteria: 6.4% (ADA), 7.9% (EASD), 22.8% (IADPSG/WHO), and 24.2% (NICE).( Nguyen CL, Lee AH, Minh Pham N, et al. Prevalence and pregnancy outcomes of gestational diabetes mellitus by different international diagnostic criteria: a prospective cohort study in Vietnam. J Matern Fetal Neonatal Med. 2020;33(21):3706-3712. doi:10.1080/14767058.2019.1583733)
Therefore, different criterion may lead to the varied materno-fetal outcomes according to Hirst et al., women with GDM by the IADPSG criterion were at risk of preterm delivery and neonatal hypoglycemia, although this criterion resulted in 20% of pregnant women being positive for GDM. The ability to cope with such a large number of cases and prevent associated adverse outcomes needs to be demonstrated before recommending widespread screening. (Hirst JE, Tran TS, Do MA, Morris JM, Jeffery HE. Consequences of gestational diabetes in an urban hospital in Viet Nam: a prospective cohort study. PLoS Med. 2012;9(7):e1001272. doi:10.1371/journal.pmed.1001272).
Due to the limitation of paper length and the aims of the study, we could not present all these points.
We only add an explained sentence in line 51-53.
-The sentence on lines 60 and 61, what is its relationship to gestational diabetes?
Authors response: The conclusion was extracted from the finding of Wei et al., the study showed a positive and significant linear relationship between FPG levels and adverse pregnancy outcomes including spontaneous abortion, PTB, macrosomia, and perinatal infant death among women without self-reported history of DM when compared with preconception FPG < 5.0 mmol/L, indicating a relatively safe FPG value for women planning to conceive in terms of preventing adverse pregnancy outcomes. As the FPG level increased per 1 mmol/L, the risk of spontaneous abortion, PTB, macrosomia, SGA, and perinatal infant death increased 8%, 3%, 5%, 3%, and 9%, respectively, using women with FPG < 5.0 mmol/L as the reference group; FPG should be used as an important evaluation indicator of preconception glycemic control when HbA1c is absent in low-resource areas.( Wei Y, Xu Q, Yang H, et al. Preconception diabetes mellitus and adverse pregnancy outcomes in over 6.4 million women: A population-based cohort study in China. PLoS Med. 2019;16(10):e1002926. Published 2019 Oct 1. doi:10.1371/journal.pmed.1002926).
The underlying mechanism is the maternal hyperglycemia and hyperinsulinemia leading to increased insulin-resistance, resulting in alterations and overnutrition in the fetus. These changes also contribute adversely to the offspring outcomes after birth. (Ornoy A, Becker M, Weinstein-Fudim L, Ergaz Z. Diabetes during Pregnancy: A Maternal Disease Complicating the Course of Pregnancy with Long-Term Deleterious Effects on the Offspring. A Clinical Review. Int J Mol Sci. 2021;22(6):2965. Published 2021 Mar 15. doi:10.3390/ijms22062965). (Murray SR, Reynolds RM. Short- and long-term outcomes of gestational diabetes and its treatment on fetal development. Prenat Diagn. 2020;40(9):1085-1091. doi:10.1002/pd.5768).
We add this explanation in line 68-70.
Materials and methods
In the materials and methods section, the authors should specify the following points:
-What was the pharmacological treatment of the participants?
Authors response: As mention in method section, if the pregnant women received an insulin therapy, they were excluded from the study. In Vietnam, metformin was not recommended for the treatment of GDM in pregnant women according to Minister of Health. Thus, nobody in this research used the oral metformin, even this medicament was approved by FDA and could be prescribed in USA.
Therefore, pharmacological treatment was not mentioned in this study.
-In which cases was the pharmacological treatment used?
Authors response: As mention above and in study flowchart, if the patients were indicated for hospitalization and required for insulin therapy, the patients were excluded from the study. Thus, the indication for pharmacological treatment was not considered in the participants.
What did the pharmacological treatment consist of?
Authors response: No drug was used in this study for the treatment of GDM. The participants could use the supplementation including iron-folic acid, zinc, calcium, vitamin D3, vitamin B12,or multiple micronutrient intake.
We add this point in line 162-163.
-Are the authors sure that their methodological design is descriptive and retrospective?
Authors response: This study enrolled all the participants visited the out-patient examination room after 2 weeks of self-management by diet therapy and physical activities. These participants had received a positive test of OGTT 2 weeks ago and under GDM counselling. At the stage of collection data, we included them and re-evaluated the management following hospital protocol. Therefore, the study was considered as a retrospective design.
-What criteria did you use to diagnose gestational diabetes?
Authors response: Thank you for this concern. We added the information in line 97-99. “This one-step screening method (75g-OGTT) was applied according to criterions of International Association of Diabetes and Pregnancy Study Groups (IADPSG) and World Health Organization (WHO).”
-Were all the participants diagnosed with gestational diabetes, or were pregnant women with carbohydrate intolerance also included?
Authors response: Thank you for this query. In line 95-96: “The participants presented at least the
occasion of the high parameter of fasting glycemia using the 75-g OGTT ≥ 92 mg/dL (≥ 5.1
mmol/L).”
In this study, we included all the pregnant women who had at least one parameter of fasting glycemia greater than 5.1 mmol/l. That means the participants were diagnosed with Gestational diabetes mellitus (GDM). The OGTT could present with one or two other negative/positive parameters (1h, 2h postprandial index). Therefore, carbohydrate intolerance was not considered in the inclusion criteria.
-What type of sampling was conducted for the selection of the 174 participants?
Authors response: Blood samples were collected from antecubital venous (line 108-109 and line 187-189).
-What does PPAQ mean? Whenever an abbreviation is used for the first time, it must be defined.
Authors response: Thank you for this correction. PPAQs was abbreviated as the self-administered Pregnancy Physical Activity Questionnaires, then median score values MET-hours/week (MET-h/week) was calculated. We added this explanation in line 131-132.
-What did the dietary and physical intervention consist of? Both interventions need to be described. Adherence to the interventions was evaluated.
Authors response: Thank you for this comment. We added the detailed description in line 147-161:
Nutrition Counselling: Three main meals including breakfast, lunch, and dinner accompanied with 2-3 snacks were required. Skipping meal was not recommended. The pregnant women were advised to use a dish of 20-cm diameter which was divided into 3 parts (Figure 2). The quarter part consists of protein (fish, carb, egg, beef, chicken, cheese, unsweetened cow’s milk, etc). The quarter part consists of carbohydrate-containing foods (rice, bread, noodle, sticky rice, rice porridge, sweet potatoes, popcorn, etc), and the half part includes vegetables and fruits. The vegetables could be cooked with oil or could be eaten fresh. Some other products such as yogurt and all kinds of beans could also supply for the carbohydrates. Saturated fat and cholesterol must be reduced in a meal.
Physical advice: The pregnant women were advised to practice with yoga, jogging, swimming, and exercises. The preconception physical activities should be continued during pregnancy. The exercises may increase from the light intensity to the moderate intensity. The time duration of physical activity should last at least 30 minutes per day. Jogging or exercises with hands in 10 minutes were recommended after meal. The pregnant women should intake enough water and maintain their exercises regularly in pregnancy.
-What criteria were used to evaluate the dietary and physical intervention at 2 weeks? It's not a very short period.
Authors response: The patients were given an instruction for diet-physical management following recommendation. After 2 weeks, we used the PPAQs to re-evaluate the result of the treatment and fasting glycemia and 2h-postprandial glycemia were collected to compare with the previous result of OGTT.
-Physical activity questionnaires during pregnancy are validated for the group of Vietnamese pregnant women.
Author response: Yes. This is an important point in our study. We previously provided in the study. In line 165-168 with a cited reference: The questionnaire survey was validated following the Vietnamese version of the pregnancy physical activity questionnaires (PPAQs) of Ota et al.
-It is not clear whether the authors only assessed the participants' usual physical activity or evaluated a physical activity programmes.
Authors response: At the first diagnosis of GDM, the patient was counselled with a nutritionist for the diet-physical management. In Vietnam, due to the limited resource settings, we managed a lot of GDM pregnant women at the same time. Thus, we did not have a specific planning for each participants. The participants should have a diet by themselves following the medical staff’s instruction and practice their free physical activities routinely such as yoga, joking, swimming, working, etc.
-If the authors only assessed the participants' usual physical activity, physical activity should not be included in the title, as it gives the impression that the participants underwent a physical activity programmes along with a medical nutrition therapy.
Authors response: Thank you for this suggestion. The usual physical activities combined with diet were recommended in hospital protocol for GDM management. Some pregnant women worried of preterm birth, thus they were intended to be stay at home or less physical activity. Thus, we need to advise them to join the routine physical activities. We have not yet a private planning for each participants. The study aims principally to investigate the GDM management with a high fasting glycemia index.
We agree to change the title to: “Non-pharmacological management of gestational diabetes mellitus with a high fasting glycemia parameter: a hospital-based study in Vietnam”
-Authors must specify all measures taken to address potential sources of bias.
Authors response: To reduce the bias, the investigators were well trained and the pregnant women were explained in detail during the survey. Moreover, the blood glucose sample was withdrawn carefully and was sent to the laboratory as rapidly as possible (less than 30 minutes). In addition, the glycemia result was analyzed using a limited number of analyzers.
We add this comment in line: 355-358.
Results
-In the results section, the authors need to improve Figure 2.
Authors response: Thank you for this comment. We separate the Figure 2 into two Figures for a better quality.
In summary, we have made substantial revisions to the manuscript to ensure that the points raised by the reviewers have been addressed thoroughly. We did not list all the changes in this letter but marked using tract change in revised paper.
We believe that these revisions have significantly improved the manuscript and hope that the reviewer and the editorial board will find the updated version satisfactory.
Once again, we would like to take this opportunity to thank you again for all your time involved and we hope the revised manuscript could be acceptable for you. We hope the revised manuscript and all the detailed responses will clarify for all your queries. We are still waiting for your decision and further instructions.
On behalf of authors,
Corresponding author,
Phuc Nhon Nguyen MD, MSc
Tu Du Hospital, Ho Chi Minh City, Vietnam
Round 2
Reviewer 2 Report
Comments and Suggestions for Authors
JCM-3082837-review-v2
The authors have satisfactorily addressed most of the observations, and at present only a few formal observations remain to be addressed before the paper can be considered for publication.
The lines referenced by the authors to the modifications made do not match the lines in the manuscript.
1. Several bibliographic citations still remain presented in separate brackets.
2. The question was quite simple; it was just connecting the following sentence to the previous text. “Other issues related to anxiety, depression, and poor mental health of pregnant women may raise a concern [12].”
3. Their methodological design is not descriptive, but rather analytical, according to the statistical analysis they carried out.
4. The question was about what type of sampling they used to recruit the participants, whether it was probabilistic or non-probabilistic
Author Response
Manuscript ID: jcm-3082837
Author's Reply to the Review Report (Reviewer 2)
The authors have satisfactorily addressed most of the observations, and at present only a few formal observations remain to be addressed before the paper can be considered for publication.
The lines referenced by the authors to the modifications made do not match the lines in the manuscript.
Authors response: Thank you for this note. The lines may be changed after the author modified the size of the figures and the tables. However, the reviewer could check the points changed by highlight track.
- Several bibliographic citations still remain presented in separate brackets.
Authors response: Thank you for this detailed correction. The authors have read carefully and amended it. Please check line 76, 101-103, 210, and line 364.
- The question was quite simple; it was just connecting the following sentence to the previous text. “Other issues related to anxiety, depression, and poor mental health of pregnant women may raise a concern [12].”
Authors response: Thank you for your suggestion. The authors added the linking word “moreover” for this point.
Moreover, other issues in associated with anxiety, depression, and poor mental health of pregnant women raise a concern in most parts of the world [21,22]. (line 76-77).
- Their methodological design is not descriptive, but rather analytical, according to the statistical analysis they carried out.
Authors response: Thank you for this correction. We change it. Please check the lines 111 and 377.
- The question was about what type of sampling they used to recruit the participants, whether it was probabilistic or non-probabilistic.
Authors response: The authors confirmed that this was a non-probabilistic sampling study (purposive selection according to study criteria).The non-probability study collected the participants upon they were presented at the examination room and met the inclusion criteria. The participants were selected until the samples were sufficient according the study sample size of 172 pregnant women at least. The study also mentioned some bias limitation of the non-random study that it may could not representative for all the Vietnamese population. Please check the line 132 and 382.
Importantly, we also would like to supply the Graphical abstract for this article please.
The authors thank again for the correction from the reviewer 2.
